# Anytime-Constrained Equilibria in Polynomial Time

**Jeremy McMahan** [1]

## Abstract

We extend anytime constraints to the Markov game setting and the corresponding solution concept of anytime-constrained equilibrium (ACE). Then, we present a comprehensive theory of anytime-constrained equilibria that includes (1) a computational characterization of feasible policies, (2) a fixed-parameter tractable algorithm for computing ACE, and (3) a polynomial-time algorithm for approximately computing ACE. Since computing a feasible policy is NP-hard even for two-player zero-sum games, our approximation guarantees are the best possible so long as $P \neq NP$. We also develop the first theory of efficient computation for action-constrained Markov games, which may be of independent interest.

## 1. Introduction

Although multi-agent reinforcement learning (MARL) has made many breakthroughs in game-playing, the literature has long since advocated the importance of constraints in real-world applications (Gu et al., 2023b). Despite their importance, the literature on constrained MARL is far behind the state-of-the-art in the single-agent setting. Most recently, almost-sure (Castellano et al., 2022) and anytime (McMahan and Zhu, 2024; McMahan, 2024) constraints have emerged in the single-agent setting to capture real-world scenarios, such as medical applications (Coronato et al., 2020; Paragliola et al., 2018; Kolesar, 1970), disaster relief scenarios (Fan et al., 2021; Wu et al., 2019; Tsai et al., 2019), and resource management (Mao et al., 2016; Li et al., 2018; Peng and Shen, 2021; Bhatia et al., 2021). However, many of these motivating applications are actually multi-agent problems. Most obviously, anytime-compliant autonomous vehicles (McMahan and Zhu, 2024; Shalev-Shwartz et al., 2016; Wu et al., 2019) must interact with other vehicles, which is a key aspect of MARL (Chu et al., 2020; Dinneweth et al.,

2022; Wiering, 2000). Despite their relevance, anytime constraints have yet to be studied in the multi-agent setting, which we remedy in this work.

Formally, we consider a constrained Markov game (cMG) $G$ with a budget vector $B$. A joint policy $\pi$ satisfies an *anytime constraint* if every player $i$'s accumulated cost is at most $B_i$ at all times: $\mathbb{P}_G^\pi[\forall h \in [H], \sum_{t=1}^{h} c_{i,t} \leq B_i] = 1$. Given such a constraint, the natural solution concept is the *anytime-constrained equilibrium* (ACE). At a high level, an ACE is a feasible joint policy $\pi$ for which no player can gain a higher value from any feasible deviation. Our main question is as follows:

> For what class of cMGs can ACE be computed (approximately) in polynomial time?

Already in the single-agent setting, McMahan and Zhu (2024) showed computing optimal anytime-constrained policies is NP-hard. The situation for games is even worse: we show that even for simple two-player zero-sum anytime-constrained MGs, computing a feasible policy is NP-hard. Furthermore, as shown in (McMahan and Zhu, 2024), expectation-constrained policies can arbitrarily violate anytime constraints, implying that standard expectation-constrained approaches fail. Moreover, even for expectation-constrained MGs, efficient algorithms are unknown outside of regret settings (Chen et al., 2022; Ding et al., 2023), which have different constraint requirements. Lastly, typical distributed learning and self-play approaches fail since the feasibility of a player's action generally depends on the choices of others.

**Past Work.** The only known efficient algorithms for anytime constraints also fail to generalize to the multi-agent setting. The approach designed in (McMahan, 2024) only applies to one constraint. On the other hand, the approach from (McMahan and Zhu, 2024) requires state augmentation and state-dependent action spaces. Although simple in the single-agent setting, the coupled nature of the players' constraints induces state-dependent action spaces that do not form product spaces. Consequently, each stage game becomes a non-normal-form game for which efficient solvers are unknown. Moreover, their approach utilizes a relaxed augmented-state space that relies on $-\infty$ to identify infeasible states. However, the non-uniqueness of equilibria in

---

[1] Department of Computer Science, University of Wisconsin-Madison, Wisconsin, USA. Correspondence to: Jeremy McMahan <jmcmahan@wisc.edu>.

*Proceedings of the 42nd International Conference on Machine Learning*, Vancouver, Canada. PMLR 267, 2025. Copyright 2025 by the author(s).

games could allow $-\infty$ solutions, which should indicate infeasibility, even when a feasible equilibrium exists.

**Our Contributions.** We present a comprehensive theory of anytime-constrained MGs, which includes (1) a computational characterization of feasible policies, (2) a fixed-parameter tractable (FPT) (Downey and Fellows, 2012) algorithm for computing subgame-perfect ACE, and (3) a polynomial time algorithm for computing approximately feasible subgame-perfect ACE. Notably, our FPT algorithm runs in polynomial time so long as the supported costs require small precision. Similarly, our approximation algorithm runs in polynomial time so long as the maximum supported cost is bounded by a polynomial factor of the budget. Given our hardness results, our algorithmic guarantees are the best possible in the worst case. Along the way, we develop efficient algorithms for constrained games, culminating in a theory of action-constrained MGs, which may be of independent interest.

Each of our main results utilizes a different algorithmic technique. For (1), we view a policy's set of feasibly realizable histories as a directed graph and then construct an AND/OR tree whose TRUE subtree is the union of all feasible policy graphs. For (2), we show that the subgame-perfect ACE of a cMG corresponds to the Markov-perfect equilibria of an action-constrained MG. Then, we design efficient algorithms for solving action-constrained games by solving a sequence of linear programs (LP). For (3), we use a combination of cost truncation and rounding to derive an approximate game whose solutions are approximately feasible equilibria for the original cMG. Then, we show that the approximate game's subgame-perfect ACE is computable in polynomial time.

### 1.1. Related Work

**Constrained MARL.** Ever since constrained equilibria were introduced (Altman and Shwartz, 2000), most works have focused on learning in the regret setting (Altman and Shwartz, 2000; Chen et al., 2022; Gattami et al., 2021; Ding et al., 2023; Jordan et al., 2024). Outside of these works, the literature has focused on single-agent-constrained Markov Decision Processes (cMDP). It is known that cMDPs can be solved in polynomial time using linear programming (Altman, 1999), and many interesting planning and learning algorithms have been developed for them (Paternain et al., 2019; Vaswani et al., 2022; Borkar, 2005; Hasanzade-Zonuzy et al., 2021). Many learning algorithms can even avoid violation during the learning process under certain assumptions (Wei et al., 2022; Bai et al., 2023). Furthermore, Brantley et al. (2020) developed no-regret algorithms for cMDPs and extended their algorithms to the setting with a constraint on the cost accumulated over all episodes, which is called a knapsack constraint (Brantley et al., 2020; Che-

ung, 2019).

**Safe MARL.** Most works implement safety using some constraints (García et al., 2015), and several multi-agent works exist down this line (Shalev-Shwartz et al., 2016; Gu et al., 2023a; Elsayed-Aly et al., 2021). The single agent setting has mainly focused on no-violation learning for cMDPs (Chow et al., 2018; Bossens and Bishop, 2022; Gu et al., 2023b) and solving CCMDPs (Wang et al., 2023; Gu et al., 2023b), which capture the probability of entering unsafe states. Performing learning while avoiding dangerous states has also been studied (Roderick et al., 2021; Thomas et al., 2021; Zhao et al., 2023) under non-trivial assumptions.

## 2. Equilibria

**Constrained Markov Games.** A (tabular, finite-horizon) $n$-player *Constrained Markov Game* (cMG) is a tuple $G = (\mathcal{S}, \mathcal{A}, P, R, C, H)$, where (i) $\mathcal{S}$ is the finite set of *states*, (ii) $\mathcal{A} = \mathcal{A}_1 \times \cdots \mathcal{A}_n$ is the finite set of *joint actions*, (iii) $P_h(s, a) \in \Delta(S)$ is the *transition* distribution, (iv) $R_h(s, a) \in \Delta(\mathbb{R}^n)$ is the *reward* distribution, (v) $C_h(s, a) \in \Delta(\mathbb{R}^n)$ is the *cost* distribution, and (vi) $H$ is the finite time *horizon*. To simplify notation, we let $r_h(s, a) \overset{\text{def}}{=} \mathbb{E}[R_h(s, a)]$ denote the expected reward, $S \overset{\text{def}}{=} |\mathcal{S}|$ denote the number of states, $A \overset{\text{def}}{=} |\mathcal{A}|$ denote the number of joint actions, $[H] \overset{\text{def}}{=} \{1, \ldots, H\}$, and $|G|$ be the total description size of the cMG.

**Interaction Protocol.** The agents interact with $G$ using a *joint policy* $\pi = \{\pi_h\}_{h=1}^H$. In the fullest generality, $\pi_h : \mathcal{H}_h \to \Delta(\mathcal{A})$ is a mapping from the observed history at time $h$ (including costs) to a distribution of actions. Often, researchers study *Markovian policies*, which take the form $\pi_h : \mathcal{S} \to \Delta(\mathcal{A})$, and *product policies*, which take the form $\pi = \{\pi_i\}_{i=1}^n$, where each $\pi_i$ is an independent policy for player $i$.

The agents start at an initial state $s_1 \in \mathcal{S}$ with observed history $\tau_1 = (s_1)$. For any $h \in [H]$, the agents choose a joint action $a_h \sim \pi_h(\tau_h)$. Then, the agents receive immediate reward vector $r_h \sim R_h(s, a)$ and cost vector $c_h \sim C_h(s, a)$. Lastly, $G$ transitions to state $s_{h+1} \sim P_h(s_h, a_h)$ and the agents update their observed history to $\tau_{h+1} = (\tau_h, a_h, c_h, s_{h+1})$. This process is repeated for $H$ steps; the interaction ends once $s_{H+1}$ is reached.

**Anytime Constraints.** Suppose the agents have a budget vector $B \in \mathbb{R}^n$. We say a joint policy $\pi$ satisfies *anytime constraints* if,

$$\mathbb{P}_G^\pi \left[ \forall h \in [H], \sum_{t=1}^h c_t \le B \right] = 1. \qquad \text{(ANY)}$$

Here, $\mathbb{P}_G^\pi$ denotes the probability law over histories induced from the interaction of $\pi$ with $G$, and all vector operations are performed component-wise. If $G$ only has anytime constraints, which will be the case in this work, we call $G$ an *anytime-constrained Markov game* (acMG). We refer to any policy $\pi$ satisfying (ANY) as *feasible* for $G$, and let $\Pi_G$ denote the set of all feasible policies for $G$.

*Remark* 2.1 (Extensions). Our results can also handle multiple constraints per agent, infinite discounting, and the weaker class of almost sure constraints. We defer the details to the appendix.

**Solution Concepts.** Solutions to games traditionally take the form of *equilibrium*. In the MARL realm, the most popular notions include the *Nash equilibrium* (NE), *correlated equilibrium* (CE), and *course-correlated equilibrium* (CCE). Given constraints, the key difference is a focus on feasible policies. Infeasible policies lead to disastrous outcomes for an agent. Thus, not only should a constrained equilibrium be feasible, but agents should only consider deviating if doing so would be feasible.

**Definition 2.2** (Anytime-Constrained Equilibria). We call a joint policy $\pi$ an *anytime-constrained equilibrium* (ACE) for an acMG $G$ if (1) $\pi \in \Pi_G$ and (2) for all players $i \in [n]$ and potential deviation policies $\pi_i'$, either,

$$(\pi_i', \pi_{-i}) \notin \Pi_G \quad \text{OR} \quad V_i^\pi \geq V_i^{\pi_i', \pi_{-i}}. \quad \text{(ACE)}$$

Here, $V_i^\pi \stackrel{\text{def}}{=} \mathbb{E}_G^\pi \left[ \sum_{t=1}^H r_{i,t} \right]$ denotes $i$'s value from interacting with $G$ under $\pi$, and $\mathbb{E}_G^\pi$ denotes the expectation defined by the law $\mathbb{P}_G^\pi$. Lastly, we call $\pi$ an *anytime-constrained Nash equilibrium* (ACNE) for $G$ if $\pi$ is additionally a product policy.

*Remark* 2.3 (Correlated Equilibrium). Our definition of ACE in Definition 2.2 technically corresponds to *anytime-constrained course-correlated equilibria* (ACCCE), which we simplify to ACE for exposition purposes. Our results apply equally well to *anytime-constrained correlated equilibria* (ACCE). We delay the definition and discussion of ACCE to the appendix. In the main text, we signal when results specialize to each equilibria type by writing ACE (NE/CE/CCE).

**Stage Games.** It is often useful to consider refinements of equilibrium notions that are more structured and robust. The classical refinement for sequential games is the *subgame-perfect equilibrium* (SPE). An SPE policy is required to behave optimally under any history, also called *subgames*, even if those subgames are not realizable. That way, players could still recover if any deviated from the policy's suggestion. In the constrained setting, we only consider feasible subgames, i.e., subgames that are realizable by some feasible policy. This means the players could still adapt and

finish the game whenever a player takes an unsupported but feasible action.

Formally, we let $\mathcal{H}_h^\pi \stackrel{\text{def}}{=} \{ \tau_h \in \mathcal{H}_h \mid \mathbb{P}_G^\pi[\tau_h] > 0 \}$ denote the subset of partial histories at time $h$ that are realizable by a policy $\pi$, and let $\mathcal{F}_h \stackrel{\text{def}}{=} \bigcup_{\pi \in \Pi_G} \mathcal{H}_h^\pi$ denote the set of partial histories at time $h$ realizable by some feasible policy. For any feasible subgame $\tau_h \in \mathcal{F}_h$, we let,

$$\Pi_G(\tau_h) \stackrel{\text{def}}{=} \left\{ \pi \mid \mathbb{P}_G^\pi \left[ \forall h \in [H], \sum_{t=1}^h c_t \leq B \mid \tau_h \right] = 1 \right\}, \quad \text{(SUB)}$$

denote the set of feasible policies for the subgame $\tau_h$. To capture our earlier intuition, we require an anytime-constrained SPE to be a policy that is feasible for any feasible subgame, and that beats any feasible deviation for that subgame.

**Definition 2.4** (Anytime-Constrained Subgame-Perfect Equilibria). We call a joint policy $\pi$ an *anytime-constrained subgame-perfect equilibrium* (ACSPE) for an acMG $G$ if for all times $h \in [H+1]$, and all feasibly-realizable histories $\tau_h \in \mathcal{F}_h$, $\pi$ satisfies (1) $\pi \in \Pi_G(\tau_h)$ and (2) for all players $i \in [n]$, and potential deviation policies $\pi_i'$, either,

$$(\pi_i', \pi_{-i}) \notin \Pi_G(\tau_h) \quad \text{OR} \quad V_{i,h}^\pi(\tau_h) \geq V_{i,h}^{\pi_i', \pi_{-i}}(\tau_h). \quad \text{(ACSPE)}$$

Here, $V_{i,h}^\pi(\tau_h) \stackrel{\text{def}}{=} \mathbb{E}_G^\pi \left[ \sum_{t=h}^H r_{i,t} \mid \tau_h \right]$ denotes $i$'s value from time $h$ onward conditioned under history $\tau_h$. Lastly, we call $\pi$ an *anytime-constrained subgame-perfect Nash equilibrium* (ACSPNE) for $G$ if $\pi$ is additionally a product policy.

Next, we show that ACSPE exists whenever a feasible policy exists. Here, we require the cost distributions to have finite support. We relax this assumption for our approximation algorithms later in Section 6.

**Assumption 2.5** (Finite Cost Support). Throughout Section 2 - Section 5, we assume each $C_h(s, a)$ has finite support.

**Proposition 2.6** (Existence). *For any acMG $G$, the following are equivalent,*

1. *$G$ admits an ACE (CE/CCE),*

2. *$G$ admits an ACSPE (CE/CCE), and*

3. *$G$ admits a feasible policy.*

*The equivalence also holds for ACNE so long as $G$ admits a feasible product policy.*

Although Proposition 2.6 implies equilibria exist under very minimal assumptions, they are generally hard to compute. As shown in (McMahan and Zhu, 2024), feasible policies

are generally not Markovian nor product policies. Moreover, just determining whether there exists a feasible policy is NP-hard even for the simplest of acMGs.

**Proposition 2.7** (Hardness). *Determining if a feasible policy exists for an acMG is NP-hard even for the restricted class of two-player zero-sum acMGs for which $S = 1$, $A = 2$, and cost functions are deterministic mappings to non-negative integers.*

# 3. Feasibility

Before we can fully understand ACE, we must first understand feasible policies. This section derives characterizations of feasibly realizable histories under anytime constraints. This leads us to design an algorithm that determines if a feasible policy exists, while also producing the set of all feasibly realizable cumulative costs and actions. These sets will be critical to our later equilibria computation in Section 4.

First, observe that for a policy $\pi$ to be feasible, all histories realizable under $\pi$ must obey the budget. If $\tau_h = (s_1, a_1, c_1, \ldots, s_h) \in \mathcal{H}_h$ is any partial history, we let $\bar{c}_h \stackrel{\text{def}}{=} \sum_{t=1}^{h-1} c_t$ denote the vector of cumulative costs induced by the history. Given this notation, we see that $\pi \in \Pi_G$ if and only if for all $h \in [H+1]$ and all $\tau_h \in \mathcal{H}_h^\pi$, it holds that $\bar{c}_h \leq B$.

**History Translation.** Consequently, we only need to consider the (state, cost)-pairs induced by a history to determine feasibility. In particular, we can focus on $\bar{\tau}_h \stackrel{\text{def}}{=} ((s_1, 0), a_1, (s_2, c_1), a_2, \ldots, (s_h, \bar{c}_h))$, which denotes $\tau_h$ written in (state, cost)-form. Observe that for any $\tau_h$, the translation of $\tau_h$ to (state, cost)-form is well-defined and unique. Specifically, given $\bar{\tau}_h$, we can infer any immediate cost $c_k$ uniquely by $c_k = \bar{c}_{k+1} - \bar{c}_k$, and given $\tau_h$, we can infer $\bar{c}_k$ uniquely by $\bar{c}_k = \sum_{t=1}^{k-1} c_t$. Given this equivalence, we focus on characterizing the following sets.

**Definition 3.1** (Feasible Sets). We define the set of state, cumulative cost pairs realizable by a feasible policy at time $h$ by,

$$\mathcal{FS}_h \stackrel{\text{def}}{=} \bigcup_{\pi \in \Pi_G} \bigcup_{\tau_h \in \mathcal{H}_h^\pi} \{(s_h, \bar{c}_h)\}. \tag{1}$$

We define the set of actions taken by some feasible policy at a pair $(s, \bar{c}) \in \mathcal{FS}_h$ by,

$$\mathcal{FA}_h(s, \bar{c}) \stackrel{\text{def}}{=} \bigcup_{\pi \in \Pi_G} \bigcup_{\substack{\tau_{h+1} \in \mathcal{H}_{h+1}^\pi, \\ (s_h, \bar{c}_h) = (s, \bar{c})}} \{a_h\}. \tag{2}$$

**Realizability Graphs.** Now, imagine that the game terminates prematurely should (1) the agents ever choose an action that could immediately violate the budget or (2) reach

a point where all available actions lead to immediate violation. Under this interpretation, it is easy to see that a policy is feasible if and only if it always reaches time $H + 1$ under any realization. We can utilize this intuition constructively through the idea of realizability graphs.

For a given policy $\pi$, we define its *realizability graph* $\mathcal{R}^\pi \stackrel{\text{def}}{=} (\mathcal{V}^\pi, \mathcal{E}^\pi)$ to be the directed acyclic multi-graph satisfying (i) $\mathcal{V}^\pi$ is the set of (time, state, cost)-triples feasibly-realizable under $\pi$, and (ii) $\mathcal{E}^\pi$ is the set of all feasible one-step (time, state, cost)-evolutions under $\pi$. We also label each edge by the action responsible for that evolution. Then, any $\pi$-realizable feasible history $\tau_h$ corresponds to the labeled path $\mathcal{P} = ((1, s_1, \bar{c}_1), a_1, \ldots, (h, s_h, \bar{c}_h))$ in $\mathcal{R}^\pi$. Thus, $\pi$ is feasible if and only if all sink nodes in $\mathcal{R}^\pi$ are distance $H$ from the source node $(1, s_1, 0)$.

**Algorithmic Approach.** Overall, we can determine if $\Pi_G \neq \varnothing$ by proving the existence of a realizability graph whose sinks are all distance $H$ from the source. Furthermore, we can compute all feasibly realizable histories by computing the union of all feasible policy realizability graphs. We accomplish this by constructing a single graph containing all feasible realizability graphs and then pruning it to match the union.

Our graph is generated by iteratively taking feasible actions. If no sequence of feasible actions ever reaches time $H + 1$, then naturally no feasible policy exists. However, we must also ensure all branches generated from an action reach time $H + 1$ to satisfy the anytime constraints. We deal with this difficulty by making each action an AND node. On the other hand, for a (time, state, cost)-triple to be feasible, there need only be a single action that ensures time $H + 1$ is reached. Thus, we make each triple an OR node. We will later show that a TRUE subgraph corresponds to our desired solution.

**Definition 3.2** (Feasibility Tree). We iteratively define an AND/OR tree $\mathcal{T}$ (Martelli and Montanari, 1973). We define the root node to be $(1, s_1, 0)$. For any time $h \in [H]$, and node $(h, s, \bar{c}) \in \mathcal{V}_\mathcal{T}$, we call $a \in \mathcal{A}$ a *feasible action* if no realization under $a$ leads to immediate violation, i.e. $\Pr_{c \sim C_h(s,a)}[\bar{c} + c \leq B] = 1$. For any feasible action $a$, we create a new AND node $u \stackrel{\text{def}}{=} (h, s, \bar{c}, a)$ and edge $(h, s, \bar{c}) \to (h, s, \bar{c}, a)$. For every $s \in \text{Supp}(P_h(s, a))$ and $c \in \text{Supp}(C_h(s, a))$, we also create a new OR node $w \stackrel{\text{def}}{=} (h + 1, s', \bar{c} + c)$ and edge $(h, s, \bar{c}, a) \to (h + 1, s', \bar{c} + c)$. We label any leaf node of the form $(H + 1, s, \bar{c})$ as TRUE and any leaf node of the form $(h, s, \bar{c})$ for $h < H + 1$ as FALSE.

Since any feasible policy can only take feasible actions by definition, any feasibly realizable history appears in (state, cost)-form as a path in $\mathcal{T}$. Moreover, conditionally feasible histories appear as superpaths in $\mathcal{T}$.

**Algorithm 1** Feasibility

**Require:** $G$
1: $\mathcal{T} \leftarrow Definition\ 3.2(G)$
2: $\text{AOSOLVE}(T)$
3: **if** $(1, s_1, 0)$ is FALSE **then**
4:     **return** "Infeasible"
5: **end if**
6: **for** $h \leftarrow 1$ to $H$ **do**
7:     $\mathcal{RS}_h \leftarrow \varnothing$ and $\mathcal{RA}_h(\cdot) \leftarrow \varnothing$
8:     **for** TRUE $(h, s, \bar{c}) \in \mathcal{V}_\mathcal{T}$ **do**
9:         $\mathcal{RS}_h \leftarrow \mathcal{RS}_h \cup \{(s, \bar{c})\}$
10:         **for** TRUE $(h, s, \bar{c}, a) \in \mathcal{E}_\mathcal{T}^+(v)$ **do**
11:             $\mathcal{RA}_h(s, \bar{c}) \leftarrow \mathcal{RA}_h(s, \bar{c}) \cup \{a\}$
12:         **end for**
13:     **end for**
14: **end for**
15: **return** $(\{\mathcal{RS}_h\}_h, \{\mathcal{RA}_h(\bar{s})\}_{h,\bar{s}})$

**Lemma 3.3.** *For any time* $h \in [H + 1]$, *and any feasibly-realizable history* $\tau_h \in \mathcal{F}_h$, *there exists a unique path* $\mathcal{P}_{\tau_h} \subseteq \mathcal{T}$ *satisfying* $\mathcal{P}_{\tau_h} = ((1, s_1, \bar{c}_1), (1, s_1, \bar{c}_1, a_1), \ldots, (h, s_h, \bar{c}_h))$. *Moreover, if* $\tau_k$ *is any suphistory of* $\tau_h$ *realizable by some* $\pi \in \Pi_G(\tau_h)$, *then* $\mathcal{P}_{\tau_h} \subseteq \mathcal{P}_{\tau_k}$.

Thus, if any feasible history exists, then there exists at least one length $H$ path in $\mathcal{T}$. However, always taking actions that are feasible at the current time is a greedy strategy that may not guarantee reaching time $H + 1$. Consequently, $\mathcal{T}$ contains many infeasible paths as well. On the bright side, we can show that any infeasible path must contain a FALSE node.

**Lemma 3.4.** *If* $\mathcal{P} = ((1, s_1, 0), (1, s_1, 0, a_1), \ldots, (h, s_h, \bar{c}_h)) \subseteq \mathcal{T}$ *is any path ending at a FALSE node, then* $\tau_h = (s_1, a_1, c_1, \ldots, s_h)$ *is not realized by any feasible policy. Moreover, if* $\tau_k$ *is any feasible subhistory of* $\tau_h$, *then* $\tau_h$ *is not realizable by any* $\pi \in \Pi_G(\tau_k)$.

**Pruning.** Overall, we see the subtree of TRUE nodes of $\mathcal{T}$ is exactly the union of all feasible policy realizability graphs. Computing the TRUE nodes for an AND/OR tree can be done in linear time using standard tree recursion (Martelli and Montanari, 1973). Suppose that AOSOLVE is any such AND/OR tree solver. Then, we can compute the $\mathcal{FS}$ and $\mathcal{FA}$ sets by implicitly pruning the FALSE nodes from $\mathcal{T}$. The full procedure is described in Algorithm 1.

**Proposition 3.5.** *For any acMG* $G$, *Algorithm 1*($G$) *outputs "Infeasible" if* $\Pi_G^B = \varnothing$ *and otherwise outputs* $(\{\mathcal{FS}_h\}_h, \{\mathcal{FA}_h(\bar{s})\}_{h,\bar{s}})$. *Moreover, Algorithm 1*($G$) *runs in time* $O((HSAD_G)^2)$, *where* $D_G \stackrel{\text{def}}{=} |\bigcup_h \bigcup_{\tau_h \in \mathcal{H}_h} \{\bar{c}_h \mid \bar{c}_h \leq B\}|$.

## 4. Reduction

As hinted in the previous section, we can convert the anytime constraint on full histories into a per-time constraint on the available actions. Specifically, if the agents track their cumulative costs, they can identify actions that satisfy the constraint long term. These actions exactly correspond to those in $\mathcal{FA}_h(s, \bar{c})$.

Then, the agents can convert their anytime-constrained MG $G$ into a traditional Markov game $\overline{G}$ with non-stationary state space $\mathcal{FS}_h$ and non-stationary, state-dependent action space $\mathcal{FA}_h(s, \bar{c})$. Importantly, $\mathcal{FA}_h(s, \bar{c})$ may induce non-normal-form subgames because the exclusion of infeasible joint actions can cause $\mathcal{FA}_h(s, \bar{c})$ not to take the form of a product space such as $\bar{\mathcal{A}}_1 \times \cdots \times \bar{\mathcal{A}}_n$. Consequently, $\overline{G}$ is an action-constrained Markov game.

**Definition 4.1.** We define an action-constrained Markov game $\overline{G} \stackrel{\text{def}}{=} (\bar{\mathcal{S}}, \bar{\mathcal{A}}, \bar{P}, \bar{R}, H)$ where,

1. $\bar{\mathcal{S}}_h \stackrel{\text{def}}{=} \mathcal{FS}_h$,

2. $\bar{\mathcal{A}}_h(s, \bar{c}) \stackrel{\text{def}}{=} \mathcal{FA}_h(s, \bar{c})$,

3. $\bar{P}_h((s', \bar{c} + c) \mid (s, \bar{c}), a) \stackrel{\text{def}}{=} C_h(c \mid s, a)P_h(s' \mid s, a)$ and,

4. $\bar{R}_h((s, \bar{c}), a) \stackrel{\text{def}}{=} R_h(s, a)$ whenever $a \in \bar{\mathcal{A}}_h(s, \bar{c})$ and $\bar{R}_h(-\infty \mid (s, \bar{c}), a) = 1$ otherwise.

In addition, we define $\overline{G}$'s initial state to be $\bar{s}_1 \stackrel{\text{def}}{=} (s_1, 0)$.

For typical MGs, the standard solution concept is the Markov-perfect equilibrium. For action-constrained MGs, we use the same solution concept, but add the condition that the policy must only support actions in the constrained action sets.

**Definition 4.2** (Markov-Perfect Equilibria). For an action-constrained MG $\overline{G}$, we say a Markovian policy $\pi$ is *all-subgame-feasible* or simply feasible in this work if $\pi_h(\bar{s}) \in \Delta(\bar{\mathcal{A}}_h(\bar{s}))$ for all times $h \in [H]$ and all states $\bar{s} \in \bar{\mathcal{S}}$. We let $\Pi_{\overline{G}}$ denote the set of all feasible policies for $\overline{G}$. Then, a Markov-perfect equilibrium (MPE) for $\overline{G}$ is a Markovian joint policy $\pi$ satisfying (1) $\pi \in \Pi_{\overline{G}}$, and (2) for all players $i \in [n]$, all times $h \in [H]$, all partial histories $\bar{\tau}_h \in \bar{H}_h$, and all potential deviation policies $\pi_i'$, either,

$$(\pi_i', \pi_{-i}) \notin \Pi_{\overline{G}} \quad \text{OR} \quad \bar{V}_{i,h}^\pi(\bar{s}_h) \geq \bar{V}_{i,h}^{\pi_i', \pi_{-i}}(\bar{\tau}_h). \tag{MPE}$$

Here, $\bar{V}_{i,h}^\pi(\bar{s}) \stackrel{\text{def}}{=} \mathbb{E}_{\overline{G}}^\pi \left[ \sum_{t=h}^H r_{i,t} \mid \bar{s}_h = \bar{s} \right]$ denotes $i$'s expected value in $\overline{G}$ under $\pi$ from time $h$ onward conditioned on starting at state $\bar{s}$. Lastly, we call $\pi$ a *Markov-perfect Nash equilibrium* (MPNE) for $\overline{G}$ if $\pi$ is additionally a product policy.

**Algorithm 2** Reduction

**Require:** $(G)$
1: $x \leftarrow Algorithm\ 1(G)$
2: **if** $x =$ "Infeasible" **then**
3:     **return** "Infeasible"
4: **end if**
5: Construct $\overline{G} \leftarrow Definition\ 4.1(G)$
6: $\pi \leftarrow \text{MGSOLVE}(\overline{G})$
7: **return** $\pi$

**Augmented Policies.** Any Markovian policy $\pi$ for $\overline{G}$ can be viewed as a history-dependent policy for $G$ represented in a compact form. In particular, by definition of $\bar{P}$, we see that the $\bar{c}$ part of $\overline{G}$'s state space always corresponds to the current cumulative cost vector. Thus, $\pi$ is equivalent to the history-dependent policy $\pi'$ formed by $\pi'_h(\tau_h) \overset{\text{def}}{=} \pi_h(s_h, \bar{c}_h)$, and can be used directly in $G$ just by feeding in the pair $(s_h, \bar{c}_h)$ to $\pi$ to generate the next joint action.

Moreover, if $\pi$ is feasible for $\overline{G}$, we see that since $\bar{\mathcal{A}}_h(s, \bar{c}) = \mathcal{F}\mathcal{A}_h(s, \bar{c})$ by definition, it must be the case that $\pi$ is feasible for $G$ by Proposition 3.5. Although less obvious, we also show that any MPE for $\overline{G}$ is an ACE for $G$.

**Lemma 4.3** (Equilibria). *Any MPE (NE/CE/CCE) for $\overline{G}$ is an ACSPE (NE/CE/CCE) for $G$.*

Then, we can compute an ACSPE for $G$ or determine that none exists by attempting to find an MPE for $\overline{G}$. We summarize the full reduction in Algorithm 2.

**Theorem 4.4** (Reduction). *For any acMG $G$, if MGSOLVE can compute a feasible MPE (NE/CE/CCE) for any feasible action-constrained Markov game, then Algorithm 2($G$) correctly outputs "Infeasible" if $\Pi_G = \varnothing$ and outputs an ACSPE (NE/CE/CCE) $\pi$, otherwise. Moreover, if MGSOLVE runs in time $O(\text{poly}(|\overline{G}|))$, then Algorithm 2($G$) runs in time $O(\text{poly}(|G|, D_G))$, and any output policy can be stored with $O(HSAD_G)$ space.*

## 5. Computation

In the last section, we showed how to reduce our anytime-constrained game problem to an action-constrained game problem. However, efficient algorithms for action-constrained games are currently unknown. In this section, we remedy this knowledge gap by designing efficient algorithms for computing MPE of an action-constrained MG. Moreover, we show that feeding our action-constrained method into our reduction yields a polynomial time algorithm for computing ACE so long as the cost precision is logarithmic.

We take a backward induction approach similar to other planning algorithms for Markov games. Unlike traditional

MGs, here, we must iteratively solve action-constrained matrix stage games. Then, we can combine the constructed policies for each stage to solve the full game. We prove the correctness of this algorithm by deriving a novel theory of equilibria in action-constrained Markov games.

**Matrix Games.** The key bottleneck to this backward induction approach is solving the action-constrained matrix games. Formally, let $(\mathcal{A}, X, u)$ denote an action-constrained matrix game, where (i) $\mathcal{A}$ is the joint action space, (ii) $X \subseteq \mathcal{A}$ is the set of feasible actions, and (iii) $u$ is the utility function. We tackle this problem by devising a variation of the standard CE/CCE LP. Importantly, we modify the constraint $\sum_{a \in \mathcal{A}} \sigma(a) = 1$, which ensures the total probability mass of all joint actions equals one, into the constraint $\sum_{a \in X} \sigma(a) = 1$, which ensures the support of the joint strategy is contained in the valid joint action space. We also define the utility of any infeasible action to be $-\infty$ so that infeasible deviations will be appropriately ignored by the LP. The full definition of the LP, which has no objective function, is,

$$\sum_{a \in X} \sigma(a)\,(u_i(a) - u_i(a'_i, a_{-i})) \geq 0, \quad \forall i, a'_i \in \mathcal{A}_i$$
$$\sum_{a \in X} \sigma(a) = 1,$$
$$\sigma(a) \geq 0 \qquad\qquad\qquad \forall a \in X$$
$$\text{(CLP)}$$

**Lemma 5.1.** *If $(\mathcal{A}, X, u)$ is any action-constrained matrix game and $\sigma$ is any solution to (CLP)$(\mathcal{A}, X, u)$, then for any player $i \in [n]$, and deviation $\sigma'_i \in \Delta(\mathcal{A}_i)$ for which $\sigma' \overset{\text{def}}{=} (\sigma'_i, \sigma_{-i}) \in \Delta(X)$, we have that $\mathbb{E}_{a \sim \sigma}[u_i(a)] \geq \mathbb{E}_{a \sim \sigma'}[u_i(a)]$. Moreover, if $X \neq \varnothing$, then there exists a solution to (CLP)$(\mathcal{A}, X, u)$.*

**Markov Games.** To use (CLP) in our backward induction, it will be useful to represent stage games with the $Q$ matrix. Formally, for any given partial policy $\pi$, any time $h \in [H]$, and any state $\bar{s} \in \bar{\mathcal{S}}$, we define the stage game to be the matrix game $\bar{Q}_h(s)$ whose utility for any player $i \in [n]$ under joint action $\bar{a} \in \bar{\mathcal{A}}$ is defined by,

$$\bar{Q}^\pi_{i,h}(\bar{s}, \bar{a}) \overset{\text{def}}{=} \begin{cases} r_h(\bar{s}, \bar{a}) + \sum_{\bar{s}'} \bar{P}_h(\bar{s}' \mid \bar{s}, \bar{a}) \bar{V}^\pi_{i,h+1}(\bar{s}') \\ -\infty \qquad \text{if } \bar{a} \notin \bar{\mathcal{A}}_h(\bar{s}) \end{cases}.$$
$$\text{(Q)}$$

Then, given some LP feasibility algorithm LPSOLVE, we use these ideas to solve action-constrained MGs in Algorithm 3.

**Theorem 5.2** (Constrained Solver). *For any feasible action-constrained MG $\overline{G}$, if LPSOLVE is a polynomial-time linear-program feasibility solver, then Algorithm 3($\overline{G}$) correctly outputs a feasible MPE (CE/CCE) in polynomial time.*

**Algorithm 3** Constrained Solver

**Require:** $\overline{G}$
1: $V_{i,H+1}^{\pi}(\bar{s}) \leftarrow 0$ for all $i \in [n]$ and $\bar{s} \in \bar{\mathcal{S}}_{H+1}$
2: **for** $h = H$ down to 1 **do**
3:    **for** $\bar{s} \in \bar{\mathcal{S}}_h$ **do**
4:       $\bar{Q}_{i,h}^{\pi}(\bar{s}, \bar{a}) \leftarrow$ (Q) for each $i \in [n]$ and $\bar{a} \in \bar{\mathcal{A}}$
5:       $\pi \leftarrow$ LPSOLVE((CLP)($\mathcal{A} = \bar{\mathcal{A}}, X = \bar{\mathcal{A}}_h(\bar{s}), u = \bar{Q}_h^{\pi}(\bar{s})))$
6:       $V_{i,h}^{\pi}(\bar{s}) \leftarrow \sum_{\bar{a}} \pi_h(\bar{a} \mid \bar{s}) Q_{i,h}^{\pi}(\bar{s}, \bar{a})$
7:    **end for**
8: **end for**
9: **return** $\pi$

**Reduction Analysis.** Given our efficient action-constrained game solver, we can now finish analyzing the running time of our reduction. Since Theorem 5.2 implies that Algorithm 3 runs in time $\mathrm{poly}(|\overline{G}|)$, Theorem 4.4 implies that Algorithm 2 with MGSOLVE = Algorithm 3 runs in time $\mathrm{poly}(|G|, D_G)$. The main issue is that $D_G$ can be exponentially large in the worst case. However, we can show that $D_G \leq \mathrm{poly}(|G|)2^{O(dn)}$, where $d$ denotes the cost precision, which is the number of significant bits needed to represent any supported cost. By definition, our method is FPT (Downey and Fellows, 2012) in the cost precision.

**Theorem 5.3** (FPT). *Equipped with any polynomial-time LP solver and* MGSOLVE = *Algorithm 3, Algorithm 2 is a fixed-parameter tractable algorithm for computing ACSPE (CE/CCE) in the cost-precision $d$. Consequently, if $d = O(\log(|G|))$ while $n$ is held constant or $d = O(1)$ while $n$ is arbitrary, then Algorithm 2 runs in polynomial time and any output policy can be stored in polynomial space.*

*Remark* 5.4 (Learning). Our methods immediately apply to the learning setting through model-based approaches. Moreover, any new learning algorithm for action-constrained MGs can immediately solve $\overline{G}$ and thus compute ACE for $G$.

# 6. Approximation

In the previous section, we showed our method runs in polynomial time whenever the cost precision is small. However, in cases where the cost precision is large or, even worse infinite, the computation may require exponential time due to the NP-hard nature of finding a feasible solution, Proposition 2.7. To combat this issue, we slightly relax the feasibility condition. This allows us to compute equilibrium policies that only violate the budget by a given $\epsilon > 0$ at the cost of an additional $\mathrm{poly}(1/\epsilon)$ factor in running time.

In this section, we allow any infinite support cost distributions that are bounded above. We also require that distribution is a product distribution to enable comparisons

between supported costs. Technically, we also need the CDF of the distribution to be efficiently computable for use in computation.

**Assumption 6.1** (Bounded). We assume that each $c^{max} \stackrel{\text{def}}{=} \sup_{h,s,a} \sup \mathrm{Supp}(C_h(s,a)) < \infty$, and that each $C_h(s,a) = \{C_{i,h}(s,a)\}_i$ is a product distribution.

Moreover, if $Hc_i^{max} \leq B$, observe that every policy is feasible for player $i$, which just leads to a standard unconstrained problem for that player. A similar phenomenon happens if $c_i^{max} \leq 0$. Thus, we assume WLOG that $Hc^{max} > B$ and $c^{max} > 0$. We can then define our relaxed feasibility notions as follows.

**Definition 6.2** (Approximate Feasibility). For any $\epsilon > 0$, a joint policy $\pi$ is $\epsilon$-additive feasible for $G$ if,

$$\mathbb{P}_G^{\pi}\left[\forall h \in [H], \sum_{t=1}^{h} c_t \leq B + \epsilon\right] = 1, \qquad (3)$$

and $\epsilon$-relative feasible for $G$ if,

$$\mathbb{P}_G^{\pi}\left[\forall h \in [H], \sum_{t=1}^{h} c_t \leq B(1 + \epsilon\sigma_B)\right] = 1, \qquad (4)$$

where $\sigma_B$ is the sign of $B$[1]. We then define an $\epsilon$-*additive/relative approximate ACE* to satisfy the usual conditions of Definition 2.2 but with the feasibility condition (1) relaxed to an $\epsilon$-additive/relative feasibility condition.

**Rounding.** The key idea of our approximations is to have players round down any cost vector they receive to the nearest multiple of some $\ell > 0$. Doing so allows the players to track far fewer cumulative costs than originally. In addition, we can effectively truncate the lower regime of the distribution using the fact that if player $i$ ever receives an immediate cost smaller than $B_i - Hc_i^{max}$, then it may take any action whatsoever going forward without violating its budget. Thus, the player can treat any smaller cost as if it were $B_i - Hc_i^{max}$. This process of rounding costs then induces a new cMG with finite support cost distributions.

**Definition 6.3** (Approximate Game). For any $\ell > 0$, we define $\lfloor c \rfloor_{\ell} \stackrel{\text{def}}{=} \lfloor \frac{c}{\ell} \rfloor \ell$ to be the largest multiple of $\ell$ that lower bounds $c$. For any player $i \in [n]$, let $\hat{c}_{i,1} \leq \cdots \leq \hat{c}_{i,m}$ denote the elements of the finite set of player $i$'s rounded costs $\{\lfloor c_i \rfloor_{\ell} \mid c_i \in [B_i - Hc_i^{max}, c_i^{max}]\}$ in order, and let $\hat{c}_{i,0} \stackrel{\text{def}}{=} -\infty$. We discretize the potentially infinite support distribution $C_{i,h}(s,a)$ into a finite support distribution $\hat{C}_{i,h}(s,a)$ by defining for each $k \in [m]$,

$$\hat{C}_{i,h}(\hat{c}_{i,k} \mid s,a) \stackrel{\text{def}}{=} \Pr_{c \sim C_{i,h}(s,a)}[c \in [\hat{c}_{i,k-1}, \hat{c}_{i,k}]]. \qquad (5)$$

---

[1]When the costs and budgets are negative, negating the constraint yields $\sum_{t=1}^{H} c_t \geq |B|(1 - \epsilon)$, which is the traditional notion of relative approximation for covering objectives.

---

**Algorithm 4** Approximation

---

**Require:** $(G)$
 1: Construct $\hat{G} \leftarrow Definition\ 6.3(G)$
 2: Let MGSOLVE = Algorithm 3
 3: **return** Algorithm $2(\hat{G})$

---

We then define the *approximate acMG* $\hat{G} \overset{\text{def}}{=} (\mathcal{S}, \mathcal{A}, P, R, \hat{C}, H)$ to be our original game $G$ but with a different cost distribution.

Since outside of extreme cases, we always round costs down, the players always maintain an underestimate of their true cumulative cost. Consequently, the players may be incurring more costs than expected. However, we can show that the true cost accumulated is not much larger than the surrogate cost.

**Lemma 6.4.** *Any feasible policy for $\hat{G}$ is an $H\ell$-additive feasible policy for $G$.*

On the flip side, the fact that players are willing to spend more than before means they have more actions available to them at each stage. Consequently, they can always find strategies that achieve the same value or even higher than any truly feasible policy for the game. It is then easy to see that solutions to $\hat{G}$ satisfy condition (2) in Definition 2.2, so form approximate ACE for $G$.

**Lemma 6.5.** *Any ACSPE (NE/CE/CCE) for $\hat{G}$ are $H\ell$-additive approximate ACSPE (NE/CE/CCE) for $G$. Moreover, if $\Pi_G \neq \varnothing$ then $\Pi_{\hat{G}} \neq \varnothing$.*

Consequently, we can compute approximate equilibria for $G$ by solving $\hat{G}$. The full algorithm is described in Algorithm 4. Since every approximate cost is an integer multiple of $\ell$, every approximate cumulative cost will also be an integer multiple of $\ell$. In fact, for each $i \in [n]$, every approximate cumulative cost's multiple must reside in the set $\left\{ H \left\lfloor \frac{B_i - Hc_i^{max}}{\ell} \right\rfloor, \ldots, H \left\lfloor \frac{c_i^{max}}{\ell} \right\rfloor \right\}$. Depending on the choice of $\ell$, the players may need to track far fewer cumulative costs to behave optimally.

**Theorem 6.6** (Approximation). *For any acMG $G$ and $\ell > 0$, Algorithm 4$(G)$ correctly outputs "Infeasible" if no $H\ell$-additive feasible policies for $G$ exist and outputs an $H\ell$-additive approximate ACSPE (CE/CCE) $\pi$, otherwise. Moreover, Algorithm 4 runs in time $O(\text{poly}(|G|, \frac{\|c^{max} - B\|_\infty^n}{\ell^n}))$.*

**Corollary 6.7** (Additive). *For any $\epsilon > 0$, if we define $\ell \overset{\text{def}}{=} \epsilon / H$, then Algorithm 4 correctly outputs "Infeasible" or an $\epsilon$-additive approximate ACSPE (CE/CCE) for any acMG. Moreover, if $\|c^{max} - B\|_\infty \leq \text{poly}(|G|)$, Algorithm 4 runs in time $O(\text{poly}(|G|, \frac{1}{\epsilon^n}))$.*

**Corollary 6.8** (Relative). *For any $\epsilon > 0$, if we define $\ell \overset{\text{def}}{=} \epsilon|B|/H$, then Algorithm 4 correctly outputs "Infeasible" or*

*an $\epsilon$-relative approximate ACSPE (CE/CCE) for any acMG. Moreover, if $c^{max} \leq \text{poly}(|G|)|B|$, Algorithm 4 runs in time $O(\text{poly}(|G|, \frac{1}{\epsilon^n}))$.*

*Remark* 6.9 (Cost Bound). We can efficiently compute approximately feasible solutions so long as $c^{max} \leq \text{poly}(|G|)|B|$. This condition is very natural. When the supported costs all have the same sign, any feasible policy induces costs with $c^{max} \leq |B|$ anyway. Importantly, this restriction is not an artifact of our approach; some bound on $c^{max}$ is necessary for efficient computation as proved in (McMahan and Zhu, 2024).

## 7. Conclusion

In this work, we introduced anytime-constraints for Markov games and studied the corresponding solution concept of anytime-constrained equilibria. Although finding a feasible policy is NP-hard for simple games, we showed efficient computation is possible so long as the cost precision is constant. The main ingredients to our approach were a graph algorithm to derive all feasibly realizable histories and an efficient algorithm for solving action-constrained MGs. Lastly, we presented approximation algorithms for computing approximately feasible anytime-constrained equilibria running in polynomial time so long as the cost distribution's supremum is no larger than a polynomial factor of the budget. Given the hardness results, our approximation guarantees are best possible under worst-case analysis.

## Impact Statement

This paper presents work whose goal is to advance the field of Machine Learning. There are many potential societal consequences of our work, none which we feel must be specifically highlighted here.

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

## A. Proofs for Section 2

### A.1. Proof of Proposition 2.6

*Proof.* If $\Pi_G = \varnothing$, then by definition no ACE or ACSPE can exist since they must be feasible. On the other hand, if $\Pi_G \neq \varnothing$, then Algorithm 2 yields an ACSPE as shown by Proposition 3.5. Thus, an ACSPE exists and so an ACE also exists. $\square$

### A.2. Proof of Proposition 2.7

*Proof.* The proof of Theorem 2 in (McMahan and Zhu, 2024) shows computing a feasible anytime-constrained policy is NP-hard for 2 constraints. Since this fact is independent of the reward structures, the result applies to two-player zero-sum cMGs by treating one player as a dummy with no influence on the transitions. $\square$

## B. Proofs for Section 3

We introduce a few helpful observations here.

**Observation 1** (Decomposability)**.** *For any policy $\pi$, time $h \in [H]$ and $\pi$-realizable partial history $\tau_{h+1} \in \mathcal{H}^{\pi}_{h+1}$, we have that $\tau_{h+1} = (\tau_h, a_h, c_h, s_{h+1})$ where,*

1. *$a_h \in Supp(\pi_h(\tau_h))$,*

2. *$c_h \in Supp(C_h(s_h, a_h))$,*

3. *$s_{h+1} \in Supp(P_h(s_h, a_h))$, and*

4. *$\mathbb{P}^{\pi}[\tau_h] > 0$.*

*Proof.* By the Markov property (Equation (2.1.11) from (Puterman, 1994)), we can decompose $\tau_{h+1} = (\tau_h, a_h, c_h, s_{h+1})$ so that,

$$\mathbb{P}^{\pi}[\tau_{h+1}] = \mathbb{P}^{\pi}[\tau_h]\pi_h(a_h \mid \tau_h)C_h(c_h \mid s_h, a_h)P_h(s_{h+1} \mid s_h, a_h). \tag{6}$$

Since $\mathbb{P}^{\pi}[\tau_{h+1}] > 0$ by assumption, it must be the case that each quantity on the RHS is also positive. In particular, we see that (i) $a_h \in \mathrm{Supp}(\pi_h(\tau_h))$, (ii) $c_h \in \mathrm{Supp}(C_h(s_h, a_h))$, (iii) $s_{h+1} \in \mathrm{Supp}(P_h(s_h, a_h))$, and (iv) $\mathbb{P}^{\pi}[\tau_h] > 0$. $\square$

**Observation 2.** *For any feasible policy $\pi \in \Pi_G$, time $h \in [H]$, and $\pi$-realizable history $\tau_h \in \mathcal{H}^{\pi}_h$, it must be that $Supp(\pi_h(\tau_h)) \subseteq \{a \in \mathcal{A} \mid \mathrm{Pr}_{c \sim C_h(s,a)}[\bar{c}_h + c \leq B] = 1\}$. The same claim holds for the set $\Pi_G(\tau_h)$ as well.*

*Proof.* Fix any such $\pi$, $h$, and $\tau_h$. Let $s \stackrel{\text{def}}{=} s_h$ and $\bar{c} \stackrel{\text{def}}{=} \bar{c}_h$. Suppose for the sake of contradiction that there exists some $a \in \mathrm{Supp}(\pi_h(\tau_h))$ satisfying $\mathrm{Pr}_{c \sim C_h(s,a)}[\bar{c} + c > B] > 0$. Then, we would have that,

$$
\begin{aligned}
\mathbb{P}^{\pi}[\exists k \in [H], \sum_{t=1}^{k} c_t > B] &\geq \mathbb{P}^{\pi}[\sum_{t=1}^{h} c_t > B] \\
&\geq \mathbb{P}^{\pi}[\sum_{t=1}^{h} c_t > B \mid \tau_h]\mathbb{P}^{\pi}[\tau_h] \\
&= \mathbb{P}^{\pi}[\bar{c} + c_h > B \mid \tau_h]\mathbb{P}^{\pi}[\tau_h] \\
&\geq \mathbb{P}^{\pi}[\tau_h]\pi_h(a \mid \tau_h) \Pr_{c \sim C_h(s,a)}[\bar{c} + c > B] \\
&> 0.
\end{aligned}
$$

The penultimate line used the Markov property, and the final line used the fact that each quantity occurs with non-zero probability by assumption. Thus, we see the existence of such an action leads to contradiction. $\square$

### B.1. Proof of Lemma 3.3

*Proof.* For the first claim, we proceed by induction on $h$.

**Base Case.** For the base case, we consider $h = 1$. In this case, the only realizable history is $\tau_1 = (s_1)$ with $\bar{c}_1 = 0$. By definition, we have $(1, s_1, 0)$ is the source node. Thus, the claim holds.

**Inductive Step.** For the inductive step, we consider any $h \geq 1$. Since $\tau_{h+1} \in \mathcal{F}_{h+1}$ by assumption, there must exist some $\pi \in \Pi_G$ that realizes $\tau_{h+1}$. Then, Observation 1 implies we can decompose $\tau_{h+1}$ into $\tau_{h+1} = (\tau_h, a_h, c_h, s_{h+1})$ satisfying conditions (i)-(iv). Since $\tau_h$ is $\pi$-realizable according to (iv), the induction hypothesis implies $\mathcal{P}_{\tau_h} = ((1, s_1, 0), (1, s_1, 0, a_1), \ldots, (h, s_h, \bar{c}_h))$ is a path in $\mathcal{T}$. Also, Observation 2 implies that $a_h$ satisfies $\Pr_{c \sim C_h(s_h, a_h)}[\bar{c}_h + c \leq B] = 1$, so $a_h$ is a feasible action for $(h, s_h, \bar{c}_h)$. By definition of $\mathcal{T}$, we then know that $(h, s_h, \bar{c}_h, a_h)$ is a node in $\mathcal{T}$ that is adjacent to $(h, s_h, \bar{c}_h)$. Similarly, by (ii) and (iii), we then know that $(h+1, s_{h+1}, \bar{c}_{h+1})$ is a node in $\mathcal{T}$ and is adjacent to $(h, s_h, \bar{c}_h, a_h)$. Thus, $\mathcal{P}_{\tau_{h+1}} = (\mathcal{P}_{\tau_h}, (h, s_h, \bar{c}_h, a_h), (h+1, s_{h+1}, \bar{c}_{h+1}))$ is the desired path in $\mathcal{T}$. This completes the induction.

For the second claim, fix any $h$ and any $\tau_h \in \mathcal{F}_h$. We show that for any $\tau_k$ that is $\pi$-realizable for some $\pi \in \Pi_G(\tau_h)$ that $\mathcal{P}_{\tau_h} \subseteq \mathcal{P}_{\tau_k}$. We proceed by induction on $k$.

**Base Case.** For the base case, we consider $k = h$. In this case, the only realizable history conditioned on $\tau_h$ is $\tau_h$ itself. Trivially, we have that $\mathcal{P}_{\tau_h} \subseteq \mathcal{P}_{\tau_h}$.

**Inductive Step.** For the inductive step, we consider any $k \geq h$. Again, we can decompose $\tau_{k+1} = (\tau_k, a_k, c_k, s_{k+1})$ by Observation 1 where $a_k$ is a feasible action for $(k, s_k, \bar{c}_k)$ by Observation 2. By the induction hypothesis, we know that $\mathcal{P}_{\tau_h} \subseteq \mathcal{P}_{\tau_k}$. Again, it is easy to see that the path $\mathcal{P}_{\tau_{k+1}} = (\mathcal{P}_{\tau_k}, (k, s_k, \bar{c}_k, a_k), (k+1, s_{k+1}, \bar{c}_{k+1}))$ is a well-defined path in $\mathcal{T}$. It is then immediate that $\mathcal{P}_{\tau_h} \subseteq \mathcal{P}_{\tau_k} \subseteq \mathcal{P}_{\tau_{k+1}}$. This completes the induction. $\square$

## B.2. Proof of Lemma 3.4

*Proof.* We then proceed by backward induction on the number of nodes in $\mathcal{P}$, $h$.

**Base Case.** For the base case, we consider $h = H + 1$. In this case, $\mathcal{P}$ ends in a $H + 1$ node, which is TRUE by definition of $\mathcal{T}$. Thus, the claim vacuously holds.

**Inductive Step.** For the inductive step, we consider any $h \leq H$. First suppose that $(h, s_h, \bar{c}_h)$ is a sink node. Then, by definition of $\mathcal{T}$, there must be no safe actions for $\tau_h$. If there were a feasible $\pi \in \Pi_G$ realizing $\tau_h$ at time $h$, then Observation 2 would imply a feasible action does exist, a contradiction. Thus, $\tau_h$ cannot be feasibly realized.

Now, suppose that $(h, s_h, \bar{c}_h)$ has outgoing edges. Since any triple-node is an OR node, we know all out-neighbors of $(h, s_h, \bar{c}_h)$ must be FALSE. In other words, for any action $a$, there exists at least one super-path $\tilde{\mathcal{P}} = (\mathcal{P}, (h, s_h, \bar{c}_h, a), (h+1, s_{h+1}, \bar{c}_{h+1}))$ ending in a FALSE node. By the induction hypothesis, we know that the corresponding history $\tau_{h+1}$ is not feasibly realizable. Therefore, any policy $\pi$ realizing $\tau_{h+1}$ must violate the budget by definition. Moreover, by definition of the edges of $\mathcal{T}$, if $\pi$ realizes $\tau_h$ and $\pi_h(a \mid \tau_h) > 0$, then $\pi$ realizes $\tau_{h+1}$. Consequently, if a policy $\pi$ realizes $\tau_h$ and $a$, then $\pi$ is not feasible. Since this holds for any possible action $a$, we have that $\tau_h$ is not feasibly realizable.

The argument for why any feasible subhistory cannot realize $\tau_h$ is nearly identical: if it could realize $\tau_h$ from some policy, then that policy must be infeasible. $\square$

## B.3. Proof of Proposition 3.5

*Proof.* The proof of correctness follows from Lemma 3.3 and Lemma 3.4. Specifically, Lemma 3.3 (along with the fact that no feasible paths are removed due to Lemma 3.4) implies that $\mathcal{RS}_h \supseteq \mathcal{FS}_h$ and $\mathcal{RA}_h(\bar{s}) \supseteq \mathcal{FA}_h(\bar{s})$ for any $\bar{s}$ since all feasibly-realizable histories appear in $\mathcal{T}$. Then, Lemma 3.4 implies that any paths containing FALSE nodes are not feasibly realizable. Consequently, $\mathcal{RS}_h \subseteq \mathcal{FS}_h$ and $\mathcal{RA}_h(\bar{s}) \subseteq \mathcal{FA}_h(\bar{s})$ for any $\bar{s}$.

For the complexity claim, we observe the time taken to construct the tree and perform a bottom-up tree evaluation is linear in the size of $\mathcal{T}$. Moreover, the final loop to construct the feasible sets also requires touching every node and edge one time. Thus, the time complexity is dominated by the size of the feasibility tree. The number of nodes in the tree is at most $HSAD_G$ since there is at most one tuple per time, state, action, and non-violating cumulative cost. Moreover, there are at

most a quadratic number of edges. Hence, the number of edges is at most $A$ times the number of nodes, leading to a total size of $O((HSAD_G)^2)$. □

# C. Proofs for Section 4

**Policy Evaluation.** For any given policy $\pi$, player $i \in [n]$, time $h \in [H+1]$, and history $\tau_h \in \mathcal{H}_h$ where $s \stackrel{\text{def}}{=} s_h$, we can compute player $i$'s value from $\pi$ under a cMG $G$ recursively using the tabular *policy evaluation equations* (Equation 4.2.6 (Puterman, 1994)). In the cMG setting, these equations take the following form:

$$V_{i,h}^\pi(\tau_h) = \mathbb{E}_{a \sim \pi_h(\tau_h)} \left[ r_{i,h}(s, a) + \sum_{c,s'} C_h(c \mid s, a) P_h(s' \mid s, a) V_{i,h+1}^\pi(\tau_h, a, c, s') \right]. \tag{CPE}$$

For a traditional or action-constrained MG $\overline{G}$, the policy evaluation equations take the more familiar form:

$$\bar{V}_{i,h}^{\bar{\pi}}(\bar{\tau}_h) = \mathbb{E}_{\bar{a} \sim \bar{\pi}_h(\tau_h)} \left[ \bar{r}_{i,h}(\bar{s}, \bar{a}) + \sum_{\bar{s}'} \bar{P}_h(\bar{s}' \mid \bar{s}, \bar{a}) \bar{V}_{i,h+1}^{\bar{\pi}}(\bar{\tau}_h, \bar{a}, \bar{s}') \right]. \tag{PE}$$

When $\bar{\pi}$ is Markovian, these equations further simplify to,

$$\bar{V}_{i,h}^{\bar{\pi}}(\bar{s}) = \mathbb{E}_{\bar{a} \sim \bar{\pi}_h(\bar{s})} \left[ \bar{r}_{i,h}(\bar{s}, \bar{a}) + \sum_{\bar{s}'} \bar{P}_h(\bar{s}' \mid \bar{s}, \bar{a}) \bar{V}_{i,h+1}^{\bar{\pi}}(\bar{s}') \right]. \tag{*PE}$$

**Policy Translations.** As mentioned in the appendix, we can immediately treat any feasible Markovian policy $\bar{\pi}$ for $\overline{G} = Definition \; 4.1(G, B)$ as a compact history-dependent policy $\pi$ for $G$ by simply using the transformation $\pi_h(\tau_h) \stackrel{\text{def}}{=} \bar{\pi}_h(s_h, \bar{c}_h)$. Going even further, we can treat history dependent policies for one game as full history dependent policies for the other. The key observation is that any history $\tau_h \in \mathcal{H}_h$ has a unique equivalent history $\bar{\tau}_h \in \bar{H}_h$ and vice versa.

Specifically, the history $\tau_h = (s_1, a_1, c_1, s_2, \ldots, s_h)$ can be effectively permuted into the history $\bar{\tau}_h = ((s_1, 0), a_1, (s_2, c_1), \ldots, (s_h, \bar{c}_h))$ and vice versa. Moreover, given $\bar{\tau}_h$ it is easy to infer any $c_k$ since $c_k = \bar{c}_{k+1} - \bar{c}_k$, and given $\tau_h$ it is easy to infer $(s_k, \bar{c}_k)$. We call $\bar{\tau}_h$ the *translation* of $\tau_h$ to $\overline{G}$ and denote it by $\bar{H}(\tau_h)$. Importantly, this conversion allows us to formally discuss a policies value in both games by simply modifying its input history.

**Lemma C.1** (Translations). *For any feasible policy $\pi \in \Pi_G$, time $h \in [H+1]$, and partial history $\tau_h \in \mathcal{H}_h^\pi$, if $\bar{\tau}_h \stackrel{\text{def}}{=} \bar{H}_h(\tau_h)$ is the translation of $\tau_h$ to $\bar{H}_h$, then $V_{i,h}^\pi(\tau_h) = \bar{V}_{i,h}^\pi(\bar{\tau}_h)$. Moreover, if $\pi$ is Markovian in $\bar{S}$, then $V_{i,h}^\pi(\tau_h) = \bar{V}_{i,h}^\pi(s_h, \bar{c}_h)$.*

*Proof.* We proceed by induction on $h$. We first note by Lemma 3.3 that all realizable histories of $\pi$ have (state, cumulative-cost)-pair in $\mathcal{FS}_h$ and actions in $\mathcal{FA}_h(\bar{s})$. Thus, in the argument below we can always assume histories realized by $\pi$ lead to a valid history for $\overline{G}$.

**Base Case.** For the base case, we consider $h = H+1$. In this case, $V_{i,H+1}^\pi(\tau_{H+1}) = 0 = \bar{V}_{i,H+1}^\pi(\bar{\tau}_{H+1})$ by definition of the value function at time $H+1$. The second claim also holds since $\bar{V}_{i,H+1}^\pi(s_{H+1}, \bar{c}_{H+1}) = 0$.

**Inductive Step.** For the inductive step, we consider any $h \leq H$. Let $s \stackrel{\text{def}}{=} s_h$ and let $\bar{s} \stackrel{\text{def}}{=} (s_h, \bar{c}_h)$. We observe by (CPE) and (PE) that,

$$V_{i,h}^{\pi}(\tau_h) = \mathbb{E}_{a \sim \pi_h(\tau_h)} \left[ r_{i,h}(s,a) + \sum_{c,s'} C_h(c \mid s,a) P_h(s' \mid s,a) V_{i,h+1}^{\pi}(\tau_h, a, c, s') \right]$$

$$= \mathbb{E}_{a \sim \pi_h(\tau_h)} \left[ r_{i,h}(s,a) + \sum_{c,s'} C_h(c \mid s,a) P_h(s' \mid s,a) \bar{V}_{i,h+1}^{\pi}(\bar{\tau}_h, a, (s', \bar{c}_h + c)) \right]$$

$$= \mathbb{E}_{a \sim \pi_h(\tau_h)} \left[ \bar{r}_{i,h}(\bar{s}, a) + \sum_{\bar{s}'} \bar{P}_h(\bar{s}' \mid \bar{s}, a) \bar{V}_{i,h+1}^{\pi}(\bar{\tau}_h, a, \bar{s}') \right]$$

$$= \mathbb{E}_{\bar{a} \sim \pi_h(\bar{\tau}_h)} \left[ \bar{r}_{i,h}(\bar{s}, \bar{a}) + \sum_{\bar{s}'} \bar{P}_h(\bar{s}' \mid \bar{s}, \bar{a}) \bar{V}_{i,h+1}^{\pi}(\bar{\tau}_h, \bar{a}, \bar{s}') \right]$$

$$= \bar{V}_{i,h}^{\pi}(\bar{\tau}_h).$$

The first line used (CPE). The second line applied the induction hypothesis along with the fact that $\tau_{h+1} = (\tau_h, a, c, s')$ translates to $\bar{\tau}_{h+1} = (\bar{\tau}_h, a, (s', \bar{c}_h + c))$ where $\bar{\tau}_h$ is the translation of $\tau_h$. The third line used the definition of $\bar{r}$ and $\bar{P}$ from Definition 4.1. The fourth line used the fact that $\pi_h(\bar{\tau}_h) = \pi_h(\tau_h)$ by definition of the translation. The last line used (PE).

For the second claim, we note if $\pi$ is Markovian in $\bar{S}$, then we can replace $\pi_h(\bar{\tau}_h)$ by $\pi_h(\bar{s})$ and inductively replace $\bar{V}_{i,h+1}^{\pi}(\bar{\tau}_h, \bar{a}, \bar{s}')$ by $\bar{V}_{i,h+1}^{\pi}(\bar{s}')$. These replacements result in the second to last line exactly matching the RHS of (*PE). Thus, $V_{i,h}^{\pi}(\tau_h) = \bar{V}_{i,h}^{\pi}(s_h, \bar{c}_h)$ in this case.

$\square$

### C.1. Proof of Lemma 4.3

We first make the following observation.

**Observation 3.** *For any policy $\pi$, if $\pi \in \Pi_{\overline{G}}$, then $\pi \in \Pi_G(\tau_h)$ for any $\tau_h \in \mathcal{F}_h$ and $h \in [H]$.*

*Proof.* This is immediate from Lemma 3.4 as any policy whose support is contained in $\bar{\mathcal{A}}_h(\bar{s}) = \mathcal{F}\mathcal{A}_h(\bar{s})$ at each stage is an anytime-feasible policy. $\square$

We will also show the following stronger claim.

**Claim 1.** *Suppose that $\pi$ is any MPE for $\overline{G}$, and that $\pi' \stackrel{def}{=} (\pi'_i, \pi_{-i}) \in \Pi_G$ is a feasible deviation for player $i$. Then, for all times $h \in [H+1]$, and all partial histories $\tau_h \in \mathcal{H}_h^{\pi'}$, we have that $V_{i,h}^{\pi}(\tau_h) \geq V_{i,h}^{\pi'}(\tau_h)$.*

*Proof.* Observe that,

$$V_{i,h}^{\pi}(\tau_h) = \bar{V}_{i,h}^{\pi}(s_h, \bar{c}_h) \geq \bar{V}_{i,h}^{\pi'}(\bar{\tau}_h) = V_{i,h}^{\pi'}(\tau_h).$$

The first equality used Lemma C.1 for a Markovian policy in $\bar{S}$. The inequality used the fact that $\pi$ is a MPE for $\overline{G}$ with $\bar{\tau}_h$ being the unique translation of $\tau_h$ to an element of $\bar{\mathcal{H}}_h$. The final equality again used Lemma C.1.

$\square$

**Proof of Lemma.** The lemma then follows as the observation yields condition (1) and the claim yields condition (2) of ACSPE.

### C.2. Proof of Theorem 4.4

*Proof.* The correctness of the algorithm is immediate from Proposition 3.5 and Lemma 4.3. For the complexity claim, the time the algorithm takes is $O((HSAD_G)^2)$ time to construct $\overline{G}$ and $\text{poly}(\overline{G})$ time to solve the LP. Since the description

size of $\overline{G}$ is also polynomial in $HSAD_G$, we then see the running time is bounded by $O(\text{poly}(|G|, D_G))$. The number of $\overline{G}$'s states is at most $O(HSD_G)$, and for each state up to $A$ joint actions' probabilities must be stored. Hence, the storage claim follows. $\qquad\square$

# D. Proofs for Section 5

## D.1. Proof of Lemma 5.1

*Proof.* Let $(\mathcal{A}, X, u)$ be any action-constrained matrix game, $\sigma$ be any solution to (CLP)$(\mathcal{A}, X, u)$, $i \in [n]$ be any player, and $\sigma'_i$ be any deviation strategy satisfying $\sigma' = (\sigma'_i, \sigma_{-i}) \in \Delta(X)$. By definition of the constraints, we see that,

$$
\begin{aligned}
\mathbb{E}_{a \sim \sigma}\left[u_i(a)\right] &= \sum_{a \in \mathcal{A}} \sigma(a)u_i(a) \\
&= \sum_{a \in X} \sigma(a)u_i(a) \\
&= \sum_{a'_i \in \mathcal{A}_i} \sigma'_i(a'_i) \sum_{a \in X} \sigma(a)u_i(a) \\
&\geq \sum_{a'_i \in \mathcal{A}_i} \sigma'_i(a'_i) \sum_{a \in X} \sigma(a)u_i(a'_i, a_{-i}) \\
&= \sum_{a'_i \in \mathcal{A}_i} \sigma'_i(a'_i) \sum_{a_{-i} \in \mathcal{A}_{-i}} \sum_{a_i \in \mathcal{A}_i} \sigma(a_i, a_{-i})u_i(a'_i, a_{-i}) \\
&= \sum_{a'_i \in \mathcal{A}_i} \sum_{a_{-i} \in \mathcal{A}_{-i}} \sigma'_i(a'_i)\sigma_{-i}(a_{-i})u_i(a'_i, a_{-i}) \\
&= \sum_{a' \in \mathcal{A}} \sigma'(a')u_i(a') \\
&= \mathbb{E}_{a' \sim \sigma'}\left[u_i(a')\right].
\end{aligned}
$$

The second line used the second constraint that ensured $\text{Supp}(\sigma) \subseteq X$. The fourth line used the first constraint. The sixth line used the definition of marginals.

For the second claim, the fact that $X \neq \varnothing$ implies there exist at least one feasible joint action, and so a feasible $\sigma$ exists. A specific $\sigma$ satisfying the other constraints is then immediate from classical game theory since it corresponds to the constraint of a normal-form game with possible $-\infty$ entries (which can be replaced by the worst possible utility minus 1). $\qquad\square$

## D.2. Proof of Theorem 5.2

We first make the following observation.

**Observation 4.** *For any feasible action-constrained MG $\overline{G}$, suppose that $\pi$ is output from Algorithm 3$(\overline{G})$. Then, $\pi \in \Pi_{\overline{G}}$.*

*Proof.* Since $\overline{G}$ is feasible, at any time $h \in [H]$ and state $\bar{s} \in \bar{\mathcal{S}}_h$, we know that $\bar{\mathcal{A}}_h(\bar{s}) \neq \varnothing$ by definition. Thus, Lemma 5.1 implies that (CLP) always outputs a solution and that solution is supported on $\bar{\mathcal{A}}_h(\bar{s})$ for any stage game $(h, \bar{s})$. Since $\pi$ is exactly the collection of all such stage solutions, we then see that $\text{Supp}(\pi_h(\bar{s})) \subseteq \bar{\mathcal{A}}_h(\bar{s})$ for all $(h, \bar{s})$. Thus, $\pi \in \Pi_{\overline{G}}$. $\qquad\square$

The following claim will also prove useful.

**Claim 2.** *For any feasible action-constrained MG $\overline{G}$, suppose that $\pi$ is output from Algorithm 3$(\overline{G})$. Then, for all players $i \in [n]$, times $h \in [H + 1]$, deviations $\pi'_i$ satisfying $\pi' \stackrel{def}{=} (\pi'_i, \pi_{-i}) \in \Pi_{\overline{G}}$, and histories $\bar{\tau}_h \in \bar{\mathcal{H}}_h^{\pi'}$, we have that $\bar{V}_{i,h}^{\pi}(\bar{s}_h) \geq \bar{V}_{i,h}^{\pi'}(\bar{\tau}_h)$.*

*Proof.* We proceed by induction on $h$.

**Base Case.** For the base case, we consider $h = H + 1$. In this case, $\bar{V}_{i,h}^{\pi}(\bar{s}) = 0 = \bar{V}_{i,h}^{\pi'}(\bar{s})$ by definition of the value function of a feasible policy at time $H + 1$.

**Inductive Step.** For the inductive step, we consider any $h \leq H$. We observe that,

$$
\begin{aligned}
\bar{V}_{i,h}^{\pi}(\bar{s}) &= \mathbb{E}_{\bar{a} \sim \pi_h(\bar{s})} \left[ \bar{r}_{i,h}(\bar{s}, \bar{a}) + \sum_{\bar{s}'} \bar{P}_h(\bar{s}' \mid \bar{s}, \bar{a}) \bar{V}_{i,h+1}^{\pi}(\bar{s}') \right] \\
&\geq \mathbb{E}_{\bar{a} \sim \pi_h(\bar{s})} \left[ \bar{r}_{i,h}(\bar{s}, \bar{a}) + \sum_{\bar{s}'} \bar{P}_h(\bar{s}' \mid \bar{s}, \bar{a}) \bar{V}_{i,h+1}^{\pi'}(\bar{\tau}_{h+1}) \right] \\
&= \mathbb{E}_{\bar{a} \sim \pi_h(\bar{s})} \left[ \bar{Q}_{i,h}^{\pi'}(\bar{\tau}_h, \bar{a}) \right] \\
&\geq \mathbb{E}_{\bar{a} \sim \pi_h'(\bar{\tau}_h)} \left[ \bar{Q}_{i,h}^{\pi'}(\bar{\tau}_h, \bar{a}) \right] \\
&= \bar{V}_{i,h}^{\pi'}(\bar{\tau}_h).
\end{aligned}
$$

The first line used (PE). The second line uses the induction hypothesis. The third line used the definition of the $Q$-function. The fourth line used Lemma 5.1 and the fact that $\text{Supp}(\pi_h(\bar{\tau}_h)') \subseteq \bar{A}_h(\bar{s}_h)$ by assumption that $\pi'$ is feasible. The last line used the relationship between the $Q$ and value functions.

$\square$

**Proof of Theorem.** By Observation 4, we know the output policy of our algorithm is feasible, and by Claim 2 we know the output policy satisfies the stage game solution condition. Thus, it is a MPE for $\overline{G}$. The running time follows since we run a polynomial time LP solver on a polynomial sized matrix game, $O(\bar{A})$, for a polynomial number of times, $O(H\bar{S})$.

### D.3. Proof of Theorem 5.3

*Proof.* We follow the same argument as in (McMahan and Zhu, 2024). By ignoring insignificant digits, we can write each number in the form $2^{-i}b_{-i} + \ldots 2^{-1}b_{-1} + 2^0 b_0 + \ldots + 2^{d-i-1}b_{d-i}$ for some $i$. By dividing by $2^{-i}$, each number is of the form $2^0 b_0 + \ldots + 2^{d-1}b_{d-1}$. Notice, the largest possible number that can be represented in this form is $\sum_{i=0}^{d-1} 2^i = 2^d - 1$. Since at each time $h$, we potentially add the maximum cost, the largest cumulative cost ever achieved is at most $2^d H - 1$. Since that is the largest cost achievable, no more than $2^d H$ can ever be achieved through all $H$ times. Similarly, no cost can be achieved smaller than $-2^d H$.

Thus, each cumulative cost is in the range $[-2^d H + 1, 2^d H - 1]$ and so at most $2^{d+1} H$ cumulative costs can ever be created. By multiplying back the $2^{-i}$ term, we see at most $2^{d+1} H$ costs are ever generated by numbers with $d$ bits of precision. Since this argument holds for each constraint independently, the total number of cumulative cost vectors that could ever be achieved is $(H 2^{d+1})^n$. Hence, $D_G \leq H^n 2^{(d+1)n}$.

Theorem 5.3 then follows immediately from Theorem 4.4, Theorem 5.2, and the definition of fixed-parameter tractability (Downey and Fellows, 2012). $\square$

## E. Proofs for Section 6

### E.1. Proof of Lemma 6.4

For any $h$ we let $\hat{c}_{h+1} := f(\tau_{h+1})$ be a random variable of the history defined inductively by $\hat{c}_1 = 0$ and $\hat{c}_{k+1} = f_k(\hat{c}_k, c_k)$ for all $k \leq h$. Here, $f$ is a function that either rounds the immediate cost or truncates to $\hat{c}_1$. Notice that since $f$ is a deterministic function, $\hat{c}_k$ can be computed from $\tau_{h+1}$ for all $k \in [h+1]$. Then, a probability distribution over $\hat{c}$ is induced by the one over histories.

*Proof.* The key observation is that for each time $h$, generally $\hat{\bar{c}}_h \leq \bar{c}_h \leq \hat{\bar{c}}_h + (h-1)\ell$ holds. The one exception is when a very negative cost is received, in which case the agents' truncation may lead to higher cost in $\hat{G}$. However, in that case, any action will still be allowed and so overestimated cost does not lead to issues. Formally, we can show

$$
\mathbb{P}_G^{\pi} \left[ \hat{\bar{c}}_h \leq \bar{c}_h \leq \hat{\bar{c}}_h + (h-1)\ell \vee \hat{\bar{c}}_h, \bar{c}_h \leq B - (H-h+1)c^{max} \right] = 1. \tag{7}
$$

The proof of this claim follows identically to the proof of Lemma 5 in (McMahan and Zhu, 2024).

Then, if $\pi$ is feasible for $\hat{G}$, we see that $\mathbb{P}^\pi \left[ \forall h \in [H], \ \hat{\bar{c}}_h \leq B \right] = 1$. Thus,

$$\mathbb{P}^\pi \left[ \forall h \in [H], \ \bar{c}_h \leq B + (h-1)\ell \right] = 1.$$

In words, $\pi$ is $H\ell$-feasible for $G$.

$\square$

### E.2. Proof of Lemma 6.5

*Proof.* Approximate feasibility of any such $\pi$ from Lemma 6.4. Moreover, we observe that there are more feasible deviations $(\pi'_i, \pi_{-i})$ in $\hat{G}$ since the cost constraint is easier to satisfy as also shown in the proof of Lemma 6.4. Thus, $\pi$ must also beat any feasible deviation for $G$. Hence, it is an approximate equilibrium. $\square$

### E.3. Proof of Theorem 6.6

*Proof.* The correctness of the algorithm follows immediately from Lemma 6.4 and Lemma 6.5. For the complexity claim, we first note that to construct $\overline{\hat{G}}$ from $\hat{G}$, we must loop over each approximate cost while performing the iteration to create $\mathcal{G}$. The number of such immediate costs per agent $i$ is the number integer multiples of $\ell$ we consider, which consists of the range $\left\{ \left\lfloor \frac{B_i - Hc_i^{max}}{\ell} \right\rfloor, \left\lfloor \frac{c_i^{max}}{\ell} \right\rfloor \right\}$. The number of elements of this set is,

$$\left\lfloor \frac{c_i^{max}}{\ell} \right\rfloor - \left\lfloor \frac{B_i - Hc_i^{max}}{\ell} \right\rfloor + 1 \leq \frac{c_i^{max}(H+1) - B_i}{\ell} + 2.$$

Thus, if we consider the worst case player, the bound becomes $\frac{\|c^{max}(H+1) - B\|_\infty}{\ell} + 2$. Since this holds independently for each player, the total number supported immediate costs in each approximate cost distribution is at most $O\left(\frac{\|c^{max}(H+1)-B\|_\infty^n}{\ell^n}\right)$. Moreover, since each immediate cost is an integer multiple of $\ell$, any cumulative cost is in the range at widest $\left\{ H \left\lfloor \frac{B_i - Hc_i^{max}}{\ell} \right\rfloor, H \left\lfloor \frac{c_i^{max}}{\ell} \right\rfloor \right\}$. Overall, we see the time needed to construct $\mathcal{G}$ and the size of the state set of $\overline{G}$ blow up by a factor of $O\left(\frac{\|c^{max}(H+1)-B\|_\infty^n}{\ell^n}\right)$. The running time claims then follow from the previous running time claims.

$\square$

### E.4. Proof of Corollary 6.7

*Proof.* The proof is immediate from Theorem 6.6 and the definition of $\ell$. $\square$

### E.5. Proof of Corollary 6.8

*Proof.* The proof is immediate from Theorem 6.6 and the definition of $\ell$. Note, here we are using a vector $\ell$. It is easy to see that the proof of Lemma 6.4 easily handles this variation. $\square$

## F. Extensions

The infinite discounted case, and generalized anytime constraints case follow similarly to in (McMahan and Zhu, 2024). As for almost sure constraints, we note that we can do the same meta-graph construction, but we start by including all possible cumulative costs, since we do not know which may eventually lead to success until we have seen the end. All other results follow similarly.

For ACCE, we observe all of our results follow with the minimal change of considering a strategic modification deviation $\phi \circ \pi$ instead of the general deviation $(\pi'_i, \pi_{-i})$ we originally considered. Our LP solution is also easily adapted by replacing the CCE condition with the CE condition.

