# OpenReview forum: "Anytime-Constrained Equilibria in Polynomial Time"
_ICML.cc/2025/Conference — ICML 2025 poster_

### Official Review · Reviewer_JvMv · 2025-03-13

**Overall Recommendation:** 3

**Summary:**

This paper introduces anytime constraints to the Markov game setting and develops a comprehensive theory of anytime-constrained equilibria (ACE). The authors present three main contributions: (1) a computational characterization of feasible policies, (2) a fixed-parameter tractable algorithm for computing ACE, and (3) a polynomial-time algorithm for approximately computing ACE. The work also develops efficient algorithms for action-constrained Markov games.

**Claims And Evidence:**

See the theoretical claim section below. The experimental evaluations are missing.

**Essential References Not Discussed:**

NA

**Experimental Designs Or Analyses:**

The experiments are not provided in the paper. It would be beneficial if authors can provide at least some basic experiments that demonstrates the usefulness of the method and compare it with the other state of the art methods.

**Methods And Evaluation Criteria:**

Equilibirum cirteria is used for the evaluation.

**Other Comments Or Suggestions:**

The paper is generally well-written and structured. The introduction provides a clear motivation for the work and situates it within the existing literature. The technical sections are detailed and rigorous. However, it is desirable to

1. More intuitive explanations or examples to illustrate key concepts
2. A dedicated section on potential applications or empirical evaluations
3. A more extensive discussion of limitations and future work directions

Overall, the paper presents a significant theoretical contribution to the field of constrained multi-agent reinforcement learning. However, it could be strengthened by including practical demonstrations and a more thorough discussion of real-world implications.

**Other Strengths And Weaknesses:**

Strengths:

See the theoretical claims section above.

Weakness:

1. Lack of experimental results or case studies
2. Limited discussion on the practical implications and potential applications
3. Assumption of finite support for cost distributions may be restrictive for some scenarios

**Questions For Authors:**

1. How does the performance of the proposed algorithms compare to existing methods in practice?
2. Can we extend the  to handle continuous state and action spaces?
3. How sensitive are the algorithms to the choice of parameters, particularly in the approximate case?
4. Are there specific real-world scenarios where this approach provides significant advantages over existing methods?

**Relation To Broader Scientific Literature:**

The paper is well written and is the first to provide anytime constraint for the mulit-agent system. The idea is therefore, novel and the  authors did a fairly good job to provide a clear disctinction the work with the existing literature.

**Theoretical Claims:**

1. Extension of anytime constraints to multi-agent reinforcement learning
2. Computational characterization of feasible policies using AND/OR trees
3. Fixed-parameter tractable algorithm for computing subgame-perfect ACE
4. Polynomial-time algorithm for approximately computing ACE
5. Development of efficient algorithms for action-constrained Markov games

---

> ### Author Rebuttal · Authors · 2025-03-31
>
> Thank you for your feedback! We will make sure to improve the paper using the suggestions mentioned. Please see our general rebuttal in the rebuttal section for Reviewer Gjeu, which addresses your concerns about empirical evaluation and existing methods. For your other concerns, please see below.
>
> **[Q1 + Q4]** These are addressed thoroughly in the general rebuttal, but we also mention here that there are no existing methods to compare to, as our work is the very first to design any algorithm whatsoever for anytime-constrained Markov games and for action-constrained Markov games.
>
> **[Q2]** Yes, the methods do extend, but the analysis is more involved, so we have deferred it to future work.
>
> **[Q3]** Which parameters is the reviewer referring to here? The only parameter in the control of the user is the violation parameter $\epsilon$. In terms of $\epsilon$, Corollary 6.7 and 6.8 indicate the running time scales as $1/\epsilon^n$ where $n$ is the number of agents. We conjecture this scaling is unavoidable due to the usual curse of multi-agents.
>
> Thanks again, and please let us know if there is anything else we can clarify!

---

### Official Review · Reviewer_7CQV · 2025-03-14

**Overall Recommendation:** 3

**Summary:**

This paper introduces the concept of anytime-constrained equilibria in the context of constrained Markov games, where agents must adhere to strict budget constraints at every time step. The authors extend the notion of anytime constraints from single-agent settings to multi-agent settings. The authors provide a method to determine whether a feasible policy exists under any time constraints. They propose an FPT algorithm for computing subgame-perfect ACE, which runs in polynomial time when the cost precision is small. At last, the authors also present an algorithm for computing approximately feasible ACE in polynomial time, provided the maximum supported cost is bounded by a polynomial factor of the budget.

**Claims And Evidence:**

Yes, all the claims are well justified.

**Essential References Not Discussed:**

N/A

**Experimental Designs Or Analyses:**

N/A

**Methods And Evaluation Criteria:**

No experiments are included in this paper.

**Other Comments Or Suggestions:**

The paper defines feasibility via realizable traces, ensuring all histories satisfy budget constraints at every step, which is ideal for safety-critical applications. An alternative is expectation constraints, where only the expected cost must stay within bounds. The authors could discuss the trade-offs between these definitions and their suitability for different applications.

**Other Strengths And Weaknesses:**

The paper introduces ACE for multi-agent systems and provides rigorous proofs for their existence, NP-hardness, and efficient computation via FPT and approximation algorithms.  However, the paper lacks empirical evaluation, which limits its practical validation, and the reliance on low-cost precision for the FPT algorithm potentially restricts scalability. The complexity is so large since $D_G$ could be significantly large.

**Questions For Authors:**

The paper assumes the existence of equilibria but does not discuss uniqueness. Are there conditions under which the anytime-constrained equilibrium is unique, and how does non-uniqueness affect the proposed algorithms?

The paper briefly mentions handling multiple constraints. How does the complexity of the algorithms scale with the number of constraints, and are there specific challenges when constraints are conflicting?

For infinite-horizon problems, how would the feasibility tree and backward induction approach need to be adapted, and what are the theoretical guarantees in this setting?

**Relation To Broader Scientific Literature:**

The paper studies cMDPs and constrained MARL, introducing concepts like ACE and providing efficient algorithms for their computation. By addressing the challenge of enforcing constraints at every time step in multi-agent settings, the paper contributes to the broader goals of safe and reliable AI, bridging the gap between theoretical research and practical applications. Its connection to prior work on approximation methods and equilibrium concepts further solidifies its relevance to the scientific literature.

**Theoretical Claims:**

The proofs look good to me. I checked the proof of main theorem 4.4.

---

> ### Author Rebuttal · Authors · 2025-03-31
>
> Thank you for your feedback! We will make sure to improve the paper using the suggestions mentioned. Please see our general rebuttal in the rebuttal section for Reviewer Gjeu, which addresses several of your concerns, including practical applicability. For your other concerns, please see below.
>
> **[Scalability]** First, we would like to emphasize that the FPT result can be restrictive if the precisions are high, but this is a fundamental limit in the problem itself rather than a flaw in our particular algorithm design. This potential scalability issue is exactly why we then derive provable approximation algorithms that run in polynomial time regardless of the precision of the input numbers. We also note that this is the very first set of algorithms for the problem, and we hope our insights will lead to faster algorithms in the future.
>
> **[Uniqueness]** To clarify, we do not assume existence in this paper. In fact, developing an algorithm to determine the existence of an ACE for an acMG is the primary purpose of section 3, which culminates in Algorithm 1. In terms of uniqueness, we do not have any characterization for the uniqueness of ACE; to our knowledge, a characterization of uniqueness is an open question for standard CCE as well. Nevertheless, the non-uniqueness of ACE is not an issue for our algorithms.
>
> **[Multiple Constraints]** Handling multiple constraints is immediate by adding dummy agents that capture that constraint. Alternatively, one could think of each agent as keeping a cumulative cost vector for each of its constraints. Either way, whether the constraints conflict or not, only affects the existence of a feasible policy, not our algorithmic approach. The complexity of our methods then scales exponentially with the number of constraints instead of the number of agents. This seems to be a fundamental bottleneck, as recent works have shown that anytime feasible policies are NP-hard to even find approximately when the number of constraints is too large, reflecting the standard curse of multi-agents.
>
> **[Infinite Discounting]** The infinite discounted setting is immediate just by using the standard trick of effective horizon. We can truncate the game at the point where the future discounted cost is at most $\epsilon/2$ for all players and then run our approximation algorithm with parameter $\epsilon/2$. This guarantees us a solution that only violates the budget by $\epsilon$ as before and retains polynomial complexity since the discounting always ensures this effective horizon will be polynomial-sized. Thus, the infinite discounted case enjoys the same theoretical claims.
>
> Thanks again, and please let us know if there is anything else we can clarify!

---

### Official Review · Reviewer_Gjeu · 2025-03-24

**Overall Recommendation:** 3

**Summary:**

The paper extends anytime constraints to the Markov game setting and the corresponding solution concept of anytime-constrained equilibrium (ACE). The authors provide: (1) a computational characterization of feasible policies, (2) a fixed-parameter tractable algorithm for computing ACE, and (3) a polynomial-time algorithm for approximately computing ACE. The approximation guarantees are the best possible unless P = NP.

**Claims And Evidence:**

The claims made in the submission are supported by clear and convincing evidence.

**Essential References Not Discussed:**

The authors have included the important references related to their work.

**Experimental Designs Or Analyses:**

N.A.

**Methods And Evaluation Criteria:**

N.A.

**Other Comments Or Suggestions:**

- Maybe it would be helpful the authors to provide a table with related settings in constrained Markov games and similar solution concepts.

**Other Strengths And Weaknesses:**

### Other strengths
- The paper is well-written and easy-to-follow.
- The paper provides an efficient computation for action-constrained Markov games, which can be of independent interest.
- The authors use interesting, and different, techniques, as well as non-trivial reductions to prove the main results.
- The main results hold for the more general settings (e.g., the infinite horizon setting)

### Weaknesses
- The anytime-constrained equilibria studied in the paper are not a well-established solution concept in the literature.

**Questions For Authors:**

- What is the relation between the anytime-constrained equilibrium and the more established constrained equilibria of [1] ?
- What can we say about Markovian/Non-Markovian and stationary/non-stationary anytime equilibria in the proposed constrained Markov game setting ?

[1] Chen, Ziyi, Shaocong Ma, and Yi Zhou. "Finding correlated equilibrium of constrained markov game: A primal-dual approach." Advances in Neural Information Processing Systems 35 (2022): 25560-25572.

**Relation To Broader Scientific Literature:**

The paper studies the solution concept of anytime-constrained equilibria in constrained Markov games. The key contributions of the paper in the context of related work are based on the definition of anytime-constrained equilibria, and essentially extend to the multi-agent setting a previous work [1], which studies single-agent constrained MDPs. The proposed constrained Markov game setting differs from the more established safe MARL setting (e.g., see [2]) introducing an extra cost function with anytime aggregated cost constraints, instead of explicitly using constraints for the value function. In the context of the proposed setting, the main results of the paper are interesting and novel.


[1] McMahan, Jeremy, and Xiaojin Zhu. "Anytime-constrained reinforcement learning." International Conference on Artificial Intelligence and Statistics. PMLR, 2024.

[2] Chen, Ziyi, Shaocong Ma, and Yi Zhou. "Finding correlated equilibrium of constrained markov game: A primal-dual approach." Advances in Neural Information Processing Systems 35 (2022): 25560-25572.

**Theoretical Claims:**

I went through the correctness of many proofs but not in depth.

---

> ### Author Rebuttal · Authors · 2025-03-31
>
> Thank you for your feedback! Please see our general rebuttal below, which addresses the relevancy of anytime constraints in the literature and comparisons to standard expectation constraints. We also provide additional commentary below the general rebuttal.
>
> --- *General Rebuttal* ---
>
> We appreciate the reviewers' thoughtful feedback and would like to address the general concerns regarding the relevancy of the ACE concept and the absence of empirical evaluation.
>
> **[Anytime Constraints Literature]** Anytime and almost-sure constraints were introduced to the Constrained Reinforcement Learning (CRL) literature to handle safety and resource management tasks and have received much recent attention. Anytime constraints are important to many applications that we mentioned in the introduction, including the popular settings:
> 1. Self-driving cars with fuel and safety constraints,
> 2. Autonomous rescue vehicle teams for disaster relief scenarios with safety and rescue constraints.
>
> Observe that both of these applications, in addition to many others, are naturally multi-agent settings, yet previous works discussing these applications have only designed algorithms for the single-agent setting. This discrepancy between the motivating applications and the work's algorithms motivated our investigation of anytime constraints in the multi-agent setting. Even though previous works have already motivated anytime-constrained multi-agent applications, we are the first to formalize and develop algorithms for these constraints in the multi-agent context. We would be happy to add more motivating applications from previous works if the reviewers feel this would be beneficial!
>
> **[Empirical Evaluation]** Being a work on theoretical foundations, we excluded empirical evaluations for two key reasons.
> 1. First, the main purpose of our work is to address the problem's intrinsic computational complexity: "For what class of cMGs can ACE be computed (approximately) in polynomial time?" This mirrors the question posed in the original paper on anytime constraints for the single-agent setting [1], allowing us to fairly compare the computational complexity of the single-agent and multi-agent settings. In particular, we showed that while exact computation is harder in the multi-agent setting, approximate computation is not much harder. Importantly, we emphasize that establishing worst-case complexity bounds requires mathematical proof; empirical evaluation cannot establish such bounds.
> 2. Second, we designed the first-ever algorithm for anytime-constrained Markov games. Consequently, there are no previous works to compare against. The most similar setting to ours is the classical expectation-constrained Markov game, but anytime constraints and expectation constraints are fundamentally different, both structurally and computationally [1]. Notably, expectation-constrained policies can arbitrarily violate anytime constraints and can be computed in polynomial time, whereas anytime-constrained policies are NP-hard to compute.
>
> Although we do not perform any empirical evaluation, we acknowledge their importance and hope our theoretical insights will be beneficial for the design of practical algorithms in the future.
>
> **[Our Contributions]** We would like to emphasize our theoretical contributions to the literature. Not only is our algorithm the *first ever* algorithm for anytime-constrained Markov games, but it is also provably *optimal* in terms of approximation guarantees. On the path to creating this algorithm, we developed new insights about anytime-constrained policies through the notion of realizability trees and established fixed-parameter tractability bounds for anytime-constrained Markov games. We also developed the first-ever polynomial-time algorithm for solving action-constrained Markov games that we use as a subroutine in our main algorithm. Our results imply the complexity of approximating anytime-constrained Markov games is at most polynomially larger than for single-agent anytime-constrained MDPs.
>
> **References**
>
> [1] "Anytime-Constrained Reinforcement Learning", Jeremy McMahan and Xiaojin Zhou. AISTATS 2024.
>
> --- *End of General Rebuttal* ---
>
> **[Q1]** As mentioned in the general rebuttal, these two constraint types are generally incomparable. In terms of applications, anytime constraints are preferred in safety contexts since an expectation-constrained policy could be arbitrarily unsafe under some realizations.
>
> **[Q2]** Generally, non-Markovian policies are not feasible for anytime constraints [1]. The policies we construct are augmented and so just a compact history dependent policy. Similarly, non-stationarity is required for finite-horizon settings [1], but stationary ACEs are possible in infinite discounted settings (though again Markovian only in the augmented space).
>
> Thank you for your feedback and let us know if you have any other questions!

---

### Decision · Program_Chairs · 2025-05-01

**Decision:**

Accept (poster)

**Comment:**

The paper introduces anytime constraints for Markov games and studies anytime-constrained equilibria. There are three main contributions: (1) a characterization of feasible policies, (2) a fixed-parameter tractable algorithm for computing anytime constrained equilibria, and (3) a polynomial-time algorithm for approximately computing anytime constrained equilibria. Overall the paper received slightly positive reviews from all the reviewers and there was an agreement about the value of the contributions of the paper. It seems the paper is above the bar for acceptance. We recommend a weak accept as there was no clear accept among the reviewers.